# Phrase Retrieval Learns Passage Retrieval, Too

**Jinhyuk Lee**                                   JINHYUKLEE@CS.PRINCETON.EDU
**Alexander Wettig**                                  AWETTIG@CS.PRINCETON.EDU
**Danqi Chen**                                        DANQIC@CS.PRINCETON.EDU
*Department of Computer Science, Princeton University*

## Abstract

Dense retrieval methods have shown great promise over sparse retrieval methods in a range of NLP problems. Among them, dense phrase retrieval—the most fine-grained retrieval unit—is appealing because phrases can be directly used as the output for question answering and slot filling tasks.[1] In this work, we follow the intuition that retrieving phrases naturally entails retrieving larger text blocks and study whether phrase retrieval can serve as the basis for coarse-level retrieval including passages and documents. We first observe that a dense phrase-retrieval system, without any retraining, already achieves better passage retrieval accuracy (+3-5% in top-5 accuracy) compared to passage retrievers, which also helps achieve superior end-to-end QA performance with fewer passages. Then, we provide an interpretation for why phrase-level supervision helps learn better fine-grained entailment compared to passage-level supervision, and also show that phrase retrieval can be improved to achieve competitive performance in document-retrieval tasks such as entity linking and knowledge-grounded dialogue. Finally, we demonstrate how phrase filtering and vector quantization can reduce the size of our index by 4-10x, making dense phrase retrieval a practical and versatile solution in multi-granularity retrieval.[2]

## 1. Introduction

Dense retrieval aims to retrieve relevant contexts from a large corpus, by learning dense representations of queries and text segments. Recently, dense retrieval of passages [Lee et al., 2019, Karpukhin et al., 2020, Xiong et al., 2021] has been shown to outperform traditional sparse retrieval methods such as TF-IDF and BM25 in a range of knowledge-intensive NLP tasks [Petroni et al., 2021], including open-domain question answering (QA) [Chen et al., 2017], entity linking [Wu et al., 2020], and knowledge-grounded dialogue [Dinan et al., 2019].

One natural design choice of these dense retrieval methods is the retrieval unit. For instance, the dense passage retriever (DPR) [Karpukhin et al., 2020] encodes a fixed-size text block of 100 words as the basic retrieval unit. On the other extreme, recent work [Seo et al., 2019, Lee et al., 2021] demonstrates that phrases can be used as a retrieval unit. In particular, Lee et al. [2021] show that learning dense representations of phrases alone can achieve competitive performance in a number of open-domain QA and slot filling tasks. This is particularly appealing since the phrases can directly serve as the output, without relying on an additional reader model to process text passages.

In this work, we draw on an intuitive motivation that every single phrase is embedded within a larger text context and ask the following question: If a retriever is able to locate

---

1. Following previous work [Seo et al., 2018, 2019], the term *phrase* denotes any contiguous text segment up to $L$ words, which is not necessarily a linguistic phrase (see Section 2).

2. Our code and models are available at `https://github.com/princeton-nlp/DensePhrases`.

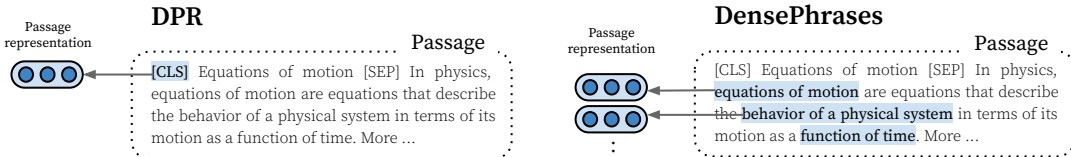

Figure 1: Comparison of passage representations from DPR [Karpukhin et al., 2020] and DensePhrases [Lee et al., 2021]. Unlike using a single vector for each passage, DensePhrases represents each passage with multiple phrase vectors and the score of a passage can be computed by the maximum score of phrases within it.

phrases, can we directly make use of it for passage and even document retrieval as well? We formulate *phrase-based passage retrieval*, in which the score of a passage is determined by the maximum score of phrases within it (see Figure 1 for an illustration). By evaluating DensePhrases [Lee et al., 2021] on popular QA datasets, we observe that it achieves competitive or even better passage retrieval accuracy compared to DPR, without any re-training or modification to the original model. The gains are especially pronounced for top-$k$ accuracy when $k$ is smaller (e.g., 5), which helps achieve strong open-domain QA accuracy with a much smaller number of passages as input to a generative model [Izacard and Grave, 2021a].

To better understand the nature of dense retrieval methods, we carefully analyze the training objectives of phrase and passage retrieval methods. While the in-batch negative losses in both models encourage them to retrieve topically relevant passages, we find that phrase-level supervision in DensePhrases provides a stronger training signal than using hard negatives from BM25, and helps DensePhrases retrieve correct phrases, and hence passages. Following this positive finding, we further explore whether phrase retrieval can be extended to retrieval of coarser granularities, or other NLP tasks. Through fine-tuning of the query encoder with document-level supervision, we are able to obtain competitive performance on entity linking [Hoffart et al., 2011] and knowledge-grounded dialogue retrieval [Dinan et al., 2019] in the KILT benchmark [Petroni et al., 2021].

Finally, we draw connections to multi-vector passage encoding models [Khattab and Zaharia, 2020, Luan et al., 2021], where phrase retrieval models can be viewed as learning a dynamic set of vectors for each passage. We show that a phrase filtering strategy learned from QA datasets gives us a control over the trade-off between the number of vectors per passage and the retrieval accuracy. Since phrase retrievers encode a larger number of vectors, we also propose a quantization-aware fine-tuning method based on Optimized Product Quantization [Ge et al., 2013], reducing the size of the phrase index from 307GB to 69GB (or under 30GB with more aggressive phrase filtering) for full English Wikipedia, without performance degradation. This matches the index size of passage retrievers and makes dense phrase retrieval a practical and versatile solution for multi-granularity retrieval.

## 2. Background

**Passage retrieval** Given a set of documents $\mathcal{D}$, passage retrieval aims to provide a set of relevant passages for a question $q$. Typically, each document in $\mathcal{D}$ is segmented into a set of disjoint passages and we denote the entire set of passages in $\mathcal{D}$ as $\mathcal{P} = \{p_1, \ldots, p_M\}$, where

each passage can be a natural paragraph or a fixed-length text block. A passage retriever is designed to return top-$k$ passages $\mathcal{P}_k \subset \mathcal{P}$ with the goal of retrieving passages that are relevant to the question. In open-domain QA, passages are considered relevant if they contain answers. However, many other knowledge-intensive NLP tasks (e.g., knowledge-grounded dialogue) provide human-annotated evidence passages or documents.

While traditional passage retrieval models rely on sparse representations such as BM25 [Robertson and Zaragoza, 2009], recent methods show promising results with dense representations of passages and questions, and enable retrieving passages that may have low lexical overlap with questions. Specifically, Karpukhin et al. [2020] introduce DPR that has a *passage encoder* $E_p(\cdot)$ and a *question encoder* $E_q(\cdot)$ trained on QA datasets and retrieves passages by using the inner product as a similarity function:

$$f(p, q) = E_p(p)^\top E_q(q). \tag{1}$$

For open-domain QA where a system is required to provide an exact answer string $a$, the retrieved top $k$ passages $\mathcal{P}_k$ are subsequently fed into a reading comprehension model such as a BERT model [Devlin et al., 2019].[3]

**Phrase retrieval** While passage retrievers require another reader model to find an answer, Seo et al. [2019] introduce the phrase retrieval approach that encodes phrases in each document and performs similarity search over all phrase vectors to directly locate the answer. Following previous work [Seo et al., 2018, 2019], we use the term 'phrase' to denote any contiguous text segment up to $L$ words (including single words), which is not necessarily a linguistic phrase and we take phrases up to length $L = 20$. Given a phrase $s^{(p)}$ from a passage $p$, their similarity function $f$ is computed as:

$$f(s^{(p)}, q) = E_s(s^{(p)})^\top E_q(q), \tag{2}$$

where $E_s(\cdot)$ and $E_q(\cdot)$ denote the *phrase encoder* and the *question encoder*, respectively. Since this formulates open-domain QA purely as a maximum inner product search (MIPS), it can drastically improve end-to-end efficiency. While previous work [Seo et al., 2019, Lee et al., 2020] relied on a combination of dense and sparse vectors, Lee et al. [2021] demonstrate that dense representations of phrases alone are sufficient to close the performance gap with retriever-reader systems. For more details on how phrase representations are learned, we refer interested readers to Lee et al. [2021].

## 3. Phrase Retrieval for Passage Retrieval

Phrases naturally have their source texts from which they are extracted. Based on this fact, we define a simple phrase-based passage retrieval strategy, where we retrieve passages based on the phrase-retrieval score:

$$\tilde{f}(p, q) := \max_{s^{(p)} \in \mathcal{S}(p)} E_s(s^{(p)})^\top E_q(q), \tag{3}$$

where $\mathcal{S}(p)$ denotes the set of phrases in the passage $p$. In practice, we first retrieve a slightly larger number of phrases, compute the score for each passage, and return top $k$

---

3. This is called the retriever-reader approach [Chen et al., 2017].

| Retriever | Natural Questions | | | | | TriviaQA | | | | |
|---|---|---|---|---|---|---|---|---|---|---|
| | Top-1 | Top-5 | Top-20 | MRR@20 | P@20 | Top-1 | Top-5 | Top-20 | MRR@20 | P@20 |
| DPR$^\diamond$ | 46.0 | 68.1 | **79.8** | 55.7 | 16.5 | 54.4$^\dagger$ | - | 79.4$^\ddagger$ | - | - |
| DPR$^\spadesuit$ | 44.2 | 66.8 | 79.2 | 54.2 | 17.7 | 54.6 | 70.8 | 79.5 | 61.7 | 30.3 |
| DensePhrases$^\diamond$ | 50.1 | 69.5 | **79.8** | 58.7 | 20.5 | - | - | - | - | - |
| DensePhrases$^\spadesuit$ | **51.1** | **69.9** | 78.7 | **59.3** | **22.7** | **62.7** | **75.0** | **80.9** | **68.2** | **38.4** |

Table 1: Open-domain QA passage retrieval results. We retrieve top $k$ passages from DensePhrases using Eq. (3). We report top-$k$ passage retrieval accuracy (Top-$k$), mean reciprocal rank at $k$ (MRR@$k$), and precision at $k$ (P@$k$). $^\diamond$: trained on each dataset independently. $^\spadesuit$: trained on multiple open-domain QA datasets. See §3.1 for more details. $^\dagger$: [Yang and Seo, 2020]. $^\ddagger$: [Karpukhin et al., 2020].

unique passages.[4] Based on our definition, phrases can act as a basic retrieval unit of any other granularity such as sentences or documents by simply changing $\mathcal{S}(p)$ (e.g., $s^{(d)} \in \mathcal{S}(d)$ for a document $d$). Note that, since the cost of score aggregation is negligible, the inference speed of phrase-based passage retrieval is the same as for phrase retrieval, which is shown to be efficient in Lee et al. [2021]. In this section, we evaluate the passage retrieval performance of phrase-based passage retrieval and how it can contribute to end-to-end open-domain QA.

## 3.1 Experiment: Passage Retrieval

**Datasets** We use two open-domain QA datasets: Natural Questions [Kwiatkowski et al., 2019] and TriviaQA [Joshi et al., 2017], following the standard train/dev/test splits for the open-domain QA evaluation. For both models, we use the 2018-12-20 Wikipedia snapshot. To provide a fair comparison, we use Wikipedia articles pre-processed for DPR, which are split into 21-million text blocks and each text block has exactly 100 words. Note that while DPR is trained in this setting, DensePhrases is trained with natural paragraphs.[5]

**Models** For DPR, we use publicly available checkpoints[6] trained on each dataset (DPR$^\diamond$) or multiple QA datasets (DPR$^\spadesuit$), which we find to perform slightly better than the ones reported in Karpukhin et al. [2020]. For DensePhrases, we train it on Natural Questions (DensePhrases$^\diamond$) or multiple QA datasets (DensePhrases$^\spadesuit$) with the code provided by the authors.[7] Note that we do not make any modification to the architecture or training methods of DensePhrases and achieve similar open-domain QA accuracy as reported. For phrase-based passage retrieval, we compute Eq. (3) with DensePhrases and return top $k$ passages.

**Metrics** Following previous work, we measure the top-$k$ passage retrieval accuracy (Top-$k$), which denotes the proportion of questions whose top $k$ retrieved passages contain at least one of the gold answers. To further characterize the behavior of each system, we include

---

4. Retrieving $2k$ phrases is often sufficient for obtaining $k$ unique passages. If not, we try $4k$ and so on.

5. We expect DensePhrases to achieve even higher performance if it is re-trained with 100-word text blocks.

6. https://github.com/facebookresearch/DPR.

7. DPR$^\spadesuit$ is trained on NaturalQuestions, TriviaQA, CuratedTREC [Baudiš and Šedivỳ, 2015], and WebQuestions [Berant et al., 2013]. DensePhrases$^\spadesuit$ additionally includes SQuAD [Rajpurkar et al., 2016], although it does not contribute to Natural Questions and TriviaQA much.

the following evaluation metrics: mean reciprocal rank at $k$ (MRR@$k$) and precision at $k$ (P@$k$). MRR@$k$ is the average reciprocal rank of the first relevant passage (that contains an answer) in the top $k$ passages. Higher MRR@$k$ means relevant passages appear at higher ranks. Meanwhile, P@$k$ is the average proportion of relevant passages in the top $k$ passages. Higher P@$k$ denotes that a larger proportion of top $k$ passages contains the answers.

**Results** As shown in Table 1, DensePhrases achieves competitive passage retrieval accuracy with DPR, while having a clear advantage on top-1 or top-5 accuracy for both Natural Questions (+6.9% Top-1) and TriviaQA (+8.1% Top-1). Although the top-20 (and top-100, which is not shown) accuracy is similar across different models, MRR@20 and P@20 reveal interesting aspects of DensePhrases—it ranks relevant passages higher and provides a larger number of correct passages. Our results suggest that DensePhrases can also retrieve passages very accurately, even though it was not explicitly trained for that purpose. For the rest of the paper, we mainly compare the DPR♠ and DensePhrases♠ models, which were both trained on multiple QA datasets.

### 3.2 Experiment: Open-domain QA

Recently, Izacard and Grave [2021a] proposed the Fusion-in-Decoder (FiD) approach where they feed top 100 passages from DPR into a generative model T5 [Raffel et al., 2020] and achieve the state-of-the-art on open-domain QA benchmarks. Since their generative model computes the hidden states of all tokens in 100 passages, it requires large GPU memory (e.g., 64 Tesla V100 32GB for training T5 [Izacard and Grave, 2021a]).

We use our phrase-based passage retrieval with DensePhrases to replace DPR in FiD and see if we can use a much smaller number of passages to achieve comparable performance, which can greatly reduce the computational requirements. We train our model with 4 24GB RTX GPUs for training T5-base,[8] which are more affordable with academic budgets.[9]

| Model | | NaturalQ | | TriviaQA |
|---|---|---|---|---|
| | | Dev | Test | Test |
| ORQA [Lee et al., 2019] | | - | 33.3 | 45.0 |
| REALM [Guu et al., 2020] | | - | 40.4 | - |
| DPR (reader: BERT-base) | | - | 41.5 | 56.8 |
| DensePhrases | | - | 41.3 | 53.5 |
| FiD with **DPR** [Izacard and Grave, 2021a] | | | | |
| Reader: T5-base | $k=5$ | 37.8 | - | - |
| | $k=10$ | 42.3 | - | - |
| | $k=25$ | 45.3 | - | - |
| | $k=50$ | 45.7 | - | - |
| | $k=100$ | 46.5 | **48.2** | **65.0** |
| FiD with **DensePhrases** (ours) | | | | |
| Reader: T5-base | $k=5$ | 44.2 | 45.9 | 59.5 |
| | $k=10$ | 45.5 | 45.9 | 61.0 |
| | $k=25$ | 46.4 | 47.2 | 63.4 |
| | $k=50$ | **47.2** | 47.9 | 64.5 |

Table 2: Open-domain QA results. We report exact match (EM) of each model by feeding top $k$ passages into a T5-base model.

**Results** As shown in Table 2, using DensePhrases as a passage retriever achieves competitive performance to DPR-based FiD and significantly improves upon the performance of original DensePhrases. Its better retrieval quality at top-$k$ for smaller $k$ indeed translates to better open-domain QA accuracy, achieving +6.4% gain compared to DPR-based FiD

---

8. Note that training T5-base with 5 or 10 passages can also be done with 11GB GPUs.

9. We also keep all the hyperparameters the same as in Izacard and Grave [2021a] except that we accumulate gradients for 16 steps to match the effective batch size of the original work.

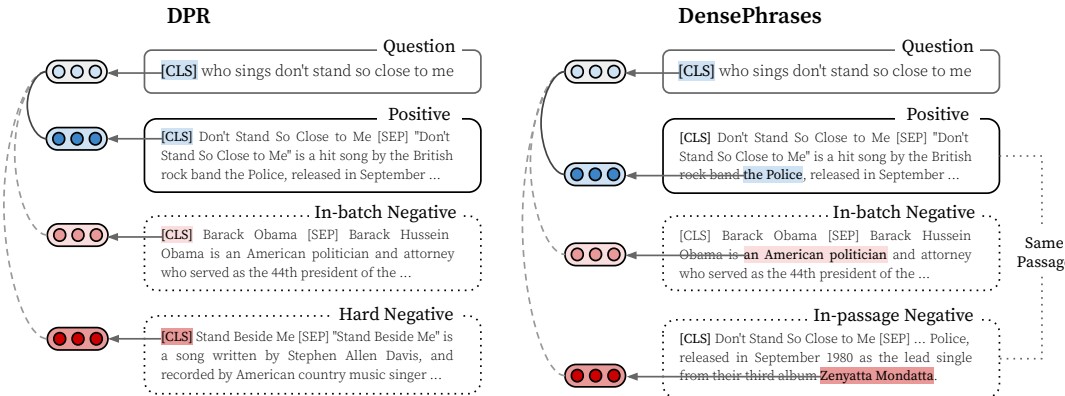

Figure 2: Comparison of training objectives of DPR and DensePhrases. While both models use in-batch negatives, DensePhrases use in-passage negatives (phrases) compared to BM25 hard-negative passages in DPR. Note that each phrase in DensePhrases can directly serve as an answer to open-domain questions.

when $k = 5$. To obtain similar performance with using 100 passages in FiD, DensePhrases needs fewer passages ($k = 25$ or $50$), which can fit in GPUs with smaller RAM.

## 4. A Unified View of Dense Retrieval

As shown in the previous section, phrase-based passage retrieval is able to achieve competitive passage retrieval accuracy, despite that the models were not explicitly trained for that. In this section, we compare the training objectives of DPR and DensePhrases in detail and explain how DensePhrases learns passage retrieval.

### 4.1 Training Objectives

Both DPR and DensePhrases set out to learn a similarity function $f$ between a passage or phrase and a question. Passages and phrases differ primarily in characteristic length, so we refer to either as a retrieval unit $x$.[10] DPR and DensePhrases both adopt a dual-encoder approach with inner product similarity as shown in Eq. (1) and (2), and they are initialized with BERT [Devlin et al., 2019] and SpanBERT [Joshi et al., 2020], respectively.

These dual-encoder models are then trained with a negative log-likelihood loss for discriminating positive retrieval units from negative ones:

$$\mathcal{L} = -\log \frac{e^{f(x^+, q)}}{e^{f(x^+, q)} + \sum\limits_{x^- \in \mathcal{X}^-} e^{f(x^-, q)}}, \tag{4}$$

where $x^+$ is the positive phrase or passage corresponding to question $q$, and $\mathcal{X}^-$ is a set of negative examples. The choice of negatives is critical in this setting and both DPR and DensePhrases make important adjustments.

**In-batch negatives** In-batch negatives are a common way to define $\mathcal{X}^-$, since they are available at no extra cost when encoding a mini-batch of examples. Specifically, in a mini-

---

10. Note that phrases may overlap, whereas passages are usually disjoint segments with each other.

batch of $B$ examples, we can add $B - 1$ in-batch negatives for each positive example. Since each mini-batch is randomly sampled from the set of all training passages, in-batch negative passages are usually *topically negative*, i.e., models can discriminate between $x^+$ and $\mathcal{X}^-$ based on their topic only.

**Hard negatives** Although topic-related features are useful in identifying broadly relevant passages, they often lack the precision to locate the exact passage containing the answer in a large corpus. Karpukhin et al. [2020] propose to use additional hard negatives which have a high BM25 lexical overlap with a given question but do not contain the answer. These hard negatives are likely to share a similar topic and encourage DPR to learn more fine-grained features to rank $x^+$ over the hard negatives. Figure 2 (left) shows an illustrating example.

**In-passage negatives** While DPR is limited to use positive passages $x^+$ which contain the answer, DensePhrases is trained to predict that the positive phrase $x^+$ *is* the answer. Thus, the fine-grained structure of phrases allows for another source of negatives, *in-passage negatives*. In particular, DensePhrases augments the set of negatives $\mathcal{X}^-$ to encompass all phrases within the same passage that do not express the answer.[11] See Figure 2 (right) for an example. We hypothesize that these in-passage negatives achieve a similar effect as DPR's hard negatives: They require the model to go beyond simple topic modeling since they share not only the same topic but also the same context. A phrase-based passage retriever might benefit from this phrase-level supervision, which has been shown to be useful in the context of distilling knowledge from reader to retriever [Izacard and Grave, 2021b].

## 4.2 Topical vs. Hard Negatives

To address our hypothesis, we would like to study how these different types of negatives used by DPR and DensePhrases affect their reliance on topical and fine-grained entailment cues. We characterize their passage retrieval based on two losses: $\mathcal{L}_{\text{topic}}$ and $\mathcal{L}_{\text{hard}}$. We use Eq. (4) to define both $\mathcal{L}_{\text{topic}}$ and $\mathcal{L}_{\text{hard}}$, but use different sets of negatives $\mathcal{X}^-$. For $\mathcal{L}_{\text{topic}}$, $\mathcal{X}^-$ contains passages that are topically different from the gold passage—In practice, we randomly sample passages from English Wikipedia. For $\mathcal{L}_{\text{hard}}$, $\mathcal{X}^-$ uses negatives containing topically similar passages, such that $\mathcal{L}_{\text{hard}}$ estimates how accurately models locate a passage that contains the exact answer among topically similar passages. From a positive passage paired with a question, we create a single hard negative by removing the sentence that contains the answer.[12] In our analysis, both metrics are estimated on the Natural Questions development set, which provides a set of questions and (gold) positive passages.

**Results** Figure 3 shows the comparison of DPR and DensePhrases trained on NQ with the two losses. For DensePhrases, we compute the passage score as described in Eq. (3). First, we observe that in-batch negatives are highly effective at reducing $\mathcal{L}_{\text{topic}}$ as DensePhrases trained with only in-passage negatives has a relatively high $\mathcal{L}_{\text{topic}}$. Furthermore, we observe that using in-passage negatives in DensePhrases (+in-passage) significantly lowers $\mathcal{L}_{\text{hard}}$,

---

11. Technically, DensePhrases treats start and end representations of phrases independently and use start (or end) representations other than the positive one as negatives.

12. While $\mathcal{L}_{\text{hard}}$ with this type of hard negatives might favor DensePhrases, using BM25 hard negatives for $\mathcal{L}_{\text{hard}}$ would favor DPR since DPR was directly trained on BM25 hard negatives. Nonetheless, we observed similar trends in $\mathcal{L}_{\text{hard}}$ regardless of the choice of hard negatives.

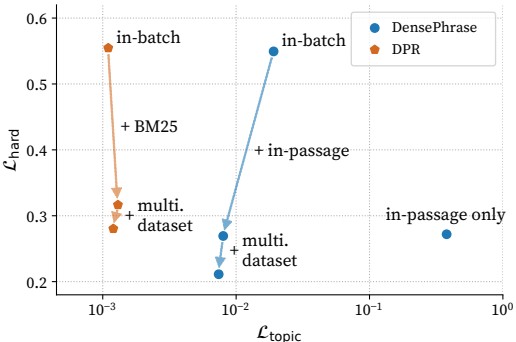

Figure 3: Analysis of DPR and DensePhrases on NQ (dev) with $\mathcal{L}_{\text{topic}}$ and $\mathcal{L}_{\text{hard}}$. Starting from a model trained with in-batch negatives (in-batch), we show the effect of using hard negatives (+BM25), in-passage negatives (+in-passage), as well as training on multiple datasets (+multi. dataset). The $x$-axis is in log-scale for better visualization. For both metrics, lower numbers are better.

| Type | $\mathcal{D} = \{p\}$ | $\mathcal{D} = \mathcal{D}_{\text{small}}$ |
|---|---|---|
| DensePhrases | 71.8 | **61.3** |
| + BM25 neg. | 71.8 | 60.6 |
| + Same-phrase neg. | **72.1** | 60.9 |

Table 3: Effect of using hard negatives in DensePhrases on the NQ development set. We report EM when a single gold passage is given ($\mathcal{D} = \{p\}$) or 6K passages are given by gathering all the gold passages from NQ development set ($\mathcal{D} = \mathcal{D}_{\text{small}}$). The two hard negatives do not give any noticeable improvement in DensePhrases.

even lower than DPR that uses BM25 hard negatives (+BM25). Using multiple datasets (+multi. dataset) further improves $\mathcal{L}_{\text{hard}}$ for both models. DPR has generally better (lower) $\mathcal{L}_{\text{topic}}$ than DensePhrases, which might be due to the smaller training batch size of DensePhrases (hence a smaller number of in-batch negatives) than DPR. The results suggest that DensePhrases relies less on topical features and is better at retrieving passages based on fine-grained entailment cues. This might contribute to the better ranking of the retrieved passages in Table 1, where DensePhrases shows better MRR@20 and P@20 while top-20 accuracy is similar.

**Hard negatives for DensePhrases?** We test two different kinds of hard negatives in DensePhrases to see whether its performance can further improve in the presence of in-passage negatives. For each training question, we mine for a hard negative passage, either by BM25 similarity or by finding another passage that contains the gold-answer phrase, but possibly with a wrong context. Then we use all phrases from the hard negative passage as additional hard negatives in $\mathcal{X}^-$ along with the existing in-passage negatives. As shown in Table 3, DensePhrases obtains no substantial improvements from additional hard negatives, indicating that in-passage negatives are already highly effective at producing good phrase (or passage) representations.

## 5. Improving Coarse-grained Retrieval

While we showed that DensePhrases implicitly learns passage retrieval, Figure 3 indicates that DensePhrases might not be very good for retrieval tasks where topic matters more than fine-grained entailment, for instance, the retrieval of a single evidence document for entity linking. In this section, we propose a simple method that can adapt DensePhrases to larger retrieval units, especially when the topical relevance is more important.

**Method** We modify the query-side fine-tuning proposed by Lee et al. [2021], which drastically improves the performance of DensePhrases by reducing the discrepancy between training and inference time. Since it is prohibitive to update the large number of phrase representations after indexing, only the query encoder is fine-tuned over the entire set of phrases in Wikipedia. Given a question $q$ and an annotated document set $\mathcal{D}^*$, we minimize:

$$\mathcal{L}_{\mathrm{doc}} = -\log \frac{\sum_{s\in\tilde{\mathcal{S}}(q),d(s)\in\mathcal{D}^*} e^{f(s,q)}}{\sum_{s\in\tilde{\mathcal{S}}(q)} e^{f(s,q)}}, \tag{5}$$

where $\tilde{\mathcal{S}}(q)$ denotes top $k$ phrases for the question $q$, out of the entire set of phrase vectors. To retrieve coarse-grained text better, we simply check the condition $d(s) \in \mathcal{D}^*$, which means $d(s)$, the source document of $s$, is included in the set of annotated gold documents $\mathcal{D}^*$ for the question. With $\mathcal{L}_{\mathrm{doc}}$, the model is trained to retrieve any phrases that are contained in a relevant document. Note that $d(s)$ can be changed to reflect any desired level of granularity such as passages.

**Datasets** We test DensePhrases trained with $\mathcal{L}_{\mathrm{doc}}$ on entity linking [Hoffart et al., 2011, Guo and Barbosa, 2018] and knowledge-grounded dialogue [Dinan et al., 2019] tasks in KILT [Petroni et al., 2021]. Entity linking contains three datasets: AIDA CoNLL-YAGO (AY2) [Hoffart et al., 2011], WNED-WIKI (WnWi) [Guo and Barbosa, 2018], and WNED-CWEB (WnCw) [Guo and Barbosa, 2018]. Each query in entity linking datasets contains a named entity marked with special tokens (i.e., [START_ENT], [END_ENT]), which need to be linked to one of the Wikipedia articles. For knowledge-grounded dialogue, we use Wizard of Wikipedia (WoW) [Dinan et al., 2019] where each query consists of conversation history, and the generated utterances should be grounded in one of the Wikipedia articles. We follow the KILT guidelines and evaluate the document (i.e., Wikipedia article) retrieval performance of our models given each query. We use R-precision, the proportion of successfully retrieved pages in the top R results, where R is the number of distinct pages in the provenance set. However, in the tasks considered, R-precision is equivalent to precision@1, since each question is annotated with only one document.

**Models** DensePhrases is trained with the original query-side fine-tuning loss (denoted as $\mathcal{L}_{\mathrm{phrase}}$) or with $\mathcal{L}_{\mathrm{doc}}$ as described in Eq. (5). When DensePhrases is trained with $\mathcal{L}_{\mathrm{phrase}}$, it labels any phrase that matches the title of gold document as positive. After training, DensePhrases returns the document that contains the top passage. For baseline retrieval methods, we report the performance of TF-IDF and DPR from Petroni et al. [2021]. We also include a multi-task version of DPR and DensePhrases, which uses the entire KILT training datasets.[13] While not our main focus of comparison, we also report the performance of other baselines from

| Model | Entity Linking | | | Dialogue |
| | AY2 | WnWi | WnCw | WoW |
|---|---|---|---|---|
| *Retriever Only* | | | | |
| TF-IDF | 3.7 | 0.2 | 2.1 | 49.0 |
| DPR | 1.8 | 0.3 | 0.5 | 25.5 |
| DensePhrases-$\mathcal{L}_{\mathrm{phrase}}$ | 7.7 | 12.5 | 6.4 | - |
| DensePhrases-$\mathcal{L}_{\mathrm{doc}}$ | 61.6 | 32.1 | 37.4 | 47.0 |
| DPR♣ | 26.5 | 4.9 | 1.9 | 41.1 |
| DensePhrases-$\mathcal{L}_{\mathrm{doc}}$♣ | 68.4 | **47.5** | **47.5** | **55.7** |
| *Retriever + Additional Components* | | | | |
| RAG | 72.6 | 48.1 | 47.6 | **57.8** |
| BLINK + flair | **81.5** | **80.2** | **68.8** | - |

Table 4: Results on the KILT test set. We report page-level R-precision on each task, which is equivalent to precision@1 on these datasets. ♣: Multi-task models.

---

13. We follow Petroni et al. [2021] for training the multi-task version of DensePhrases.

Petroni et al. [2021], which uses generative models (e.g., RAG [Lewis et al., 2020]) or task-specific models (e.g., BLINK [Wu et al., 2020], which has additional entity linking pre-training). Note that these methods use additional components such as a generative model or a cross-encoder model on top of retrieval models.

**Results** Table 4 shows the results on entity linking and knowledge-grounded dialogue tasks. On all tasks, we find that DensePhrases-$\mathcal{L}_{\mathrm{doc}}$ performs much better than DensePhrases-$\mathcal{L}_{\mathrm{phrase}}$ and also matches the performance of RAG that uses an additional large generative model. Using $\mathcal{L}_{\mathrm{phrase}}$ does very poorly since it focuses on phrase-level entailment, rather than document-level relevance. Compared to the multi-task version of DPR (i.e., DPR♣), DensePhrases-$\mathcal{L}_{\mathrm{doc}}$♣ can be easily adapted to non-QA tasks like entity linking and generalizes better on tasks without training sets (WnWi, WnCw).

## 6. DensePhrases as a Multi-Vector Passage Encoder

In this section, we demonstrate that DensePhrases can be interpreted as a multi-vector passage encoder, which has recently been shown to be very effective for passage retrieval [Luan et al., 2021, Khattab and Zaharia, 2020]. Since this type of multi-vector encoding models requires a large disk footprint, we show that we can control the number of vectors per passage (and hence the index size) through filtering. We also introduce quantization techniques to build more efficient phrase retrieval models without a significant performance drop.

### 6.1 Multi-Vector Encodings

Since we represent passages not by a single vector, but by a set of phrase vectors (decomposed as token-level start and end vectors, see Lee et al. [2021]), we notice similarities to previous work, which addresses the capacity limitations of dense, fixed-length passage encodings. While these approaches store a fixed number of vectors per passage [Luan et al., 2021, Humeau et al., 2020] or all token-level vectors [Khattab and Zaharia, 2020], phrase retrieval models store a dynamic number of phrase vectors per passage, where many phrases are filtered by a model trained on QA datasets. Specifically, Lee et al. [2021] trains a binary classifier (or a phrase filter) to filter phrases based on their phrase representations. This phrase filter is supervised by the answer annotations in QA datasets, hence denotes candidate answer phrases. In our experiment, we tune the filter threshold to control the number of vectors per passage for passage retrieval.

### 6.2 Efficient Phrase Retrieval

The multi-vector encoding models as well as ours are prohibitively large since they contain multiple vector representations for every passage in the entire corpus. We introduce a vector quantization-based method that can safely reduce the size of our phrase index, without performance degradation.

**Optimized product quantization** Since the multi-vector encoding models are prohibitively large due to their multiple representations, we further introduce a vector quantization-based method that can safely reduce the size of our phrase index, without performance degradation. We use Product Quantization (PQ) [Jegou et al., 2010] where the original

vector space is decomposed into the Cartesian product of subspaces. Among different variants of PQ, we use Optimized Product Quantization (OPQ) [Ge et al., 2013], which learns an orthogonal matrix $R$ to better decompose the original vector space. See Ge et al. [2013] for more details on OPQ.

**Quantization-aware training** While this type of aggressive vector quantization can significantly reduce memory usage, it often comes at the cost of performance degradation due to the quantization loss. To mitigate this problem, we use quantization-aware query-side fine-tuning motivated by quantization-aware training [Jacob et al., 2018]. Specifically, during query-side fine-tuning, we reconstruct the phrase vectors using the trained (optimized) product quantizer, which are then used to minimize Eq. (5).

### 6.3 Experimental Results

In Figure 4, we present the top-5 passage retrieval accuracy with respect to the size of the phrase index in DensePhrases. First, applying OPQ can reduce the index size of DensePhrases from 307GB to 69GB, while the top-5 retrieval accuracy is poor without quantization-aware query-side fine-tuning. Furthermore, by tuning the threshold $\tau$ for the phrase filter, the number of vectors per each passage (# vec / p) can be reduced without hurting the performance significantly. The performance improves with a larger number of vectors per passage, which aligns with the findings of multi-vector encoding models [Khattab and Zaharia, 2020, Luan et al., 2021]. Our results show that having 8.8 vectors per passage in DensePhrases has similar retrieval accuracy with DPR.

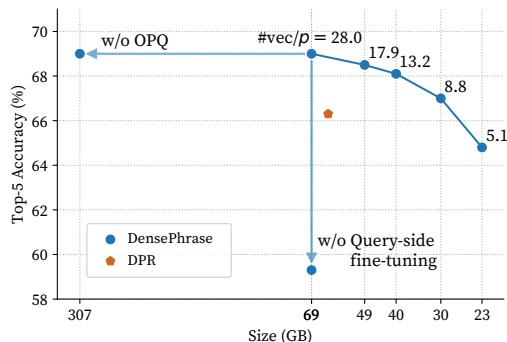

Figure 4: Top-5 passage retrieval accuracy on Natural Questions (dev) for different index sizes of DensePhrases. The index size (GB) and the average number of saved vectors per passage (# vec / p) are controlled by the filtering threshold $\tau$. For instance, # vec / p reduces from 28.0 to 5.1 with higher $\tau$, reducing the index size from 69GB to 23GB. OPQ: Optimized Product Quantization [Ge et al., 2013].

## 7. Conclusion

In this paper, we show that phrase retrieval models also learn passage retrieval without any modification. By drawing connections between the objectives of DPR and DensePhrases, we provide a better understanding of how phrase retrieval learns passage retrieval, which is also supported by several empirical evaluations on multiple benchmarks. Specifically, phrase-based passage retrieval has better retrieval quality on top $k$ passages when $k$ is small, and this translates to an efficient use of passages for open-domain QA. We also show that DensePhrases can be fine-tuned for more coarse-grained retrieval units, serving as a basis for any retrieval unit. We plan to further evaluate phrase-based passage retrieval on standard information retrieval tasks such as MS MARCO.

## Acknowledgements

We thank Chris Sciavolino, Xingcheng Yao, the members of the Princeton NLP group, and the anonymous reviewers for helpful discussion and valuable feedback. This research is supported by the James Mi *91 Research Innovation Fund for Data Science and gifts from Apple and Amazon. It was also supported in part by the ICT Creative Consilience program (IITP-2021-0-01819) supervised by the IITP (Institute for Information & communications Technology Planning & Evaluation) and National Research Foundation of Korea (NRF-2020R1A2C3010638).

## Ethical Considerations

Models introduced in our work often use question answering datasets such as Natural Questions to build phrase or passage representations. Some of the datasets, like SQuAD, are created from a small number of popular Wikipedia articles, hence could make our model biased towards a small number of topics. We hope that inventing an alternative training method that properly regularizes our model could mitigate this problem. Although our efforts have been made to reduce the computational cost of retrieval models, using passage retrieval models as external knowledge bases will inevitably increase the resource requirements for future experiments. Further efforts should be made to make retrieval more affordable for independent researchers.

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
