# OpenReview forum: "Phrase Retrieval Learns Passage Retrieval, Too"
_AKBC.ws/2021/Conference — AKBC 2021_

### Official Review · Reviewer_C1RY · 2021-07-19
**A comprehensive and thorough study of the utility of dense phrase retrieval for larger-context retrieval**

**Rating:** 7
**Confidence:** 3

**Review:**

### Summary

The paper investigates the utility of dense phrase retrieval for retrieval of passages and documents, based on the intuition that every phrase is derived from a larger context. The authors show that (a) scoring a passage based on the maximal phrase score of all phrases in the passage improves retrieval performance over passage retrieval models (DPR), and (b) that this improvement translates to better QA accuracy on two open-domain QA datasets (Natural Questions and TriviaQA). Further analysis highlights the differences between models trained for passage retrieval and phrase-based passage retrieval, proposing that DensePhrases struggles with retrieval tasks where topic matters while it is better at fine-grained entailment. The authors then propose a variant of DensePhrases adapted for larger contexts, and measure its performance on entity-linking and dialogue tasks from the KILT benchmark, showing a significant boost in performance compared to using the original DensePhrases retrieval. Yet, it looks like phrase-based retrieval is far behind the RAG baseline in terms of R-precision, which (unless I missed it) is not mentioned in the paper. Last, the authors draw the connection between their method and recently published work about multi-vector passage encoding, and demonstrate the tradeoff between retrieval performance and the number of vectors per passage, through a simple filtering method. By employing this filtering and using quantization in fine-tuning, they substantially reduce the size of the index, which makes this method more efficient at almost no cost in retrieval performance.


### Strengths

The paper is very well written and rich in experiments and interesting findings, which provide a clear and comprehensive view of the capabilities and limitations of using DensePhrases to retrieve larger contexts.

The paper is timely and draws meaningful connections to recent related work.



### Weaknesses

Results on the KILT tasks (Table 3) are not fully clear/convincing. The DensePhrases variant ($\mathcal{L}_{\text{doc}}$) gives a significant boost in performance compared to the original DensePhrases, but it doesn't seem to perform comparably well with RAG in terms of R-precision. Actually, the RAG baseline is not mentioned anywhere in the text and the corresponding paper is not cited.

The Topical vs. Hard Negatives analysis (section 4.2, figure 3) is missing a baseline. The choice of passages for $\mathcal{L}_{\text{hard}}$ makes sense (using the original hard negative BM-25 based examples of DPR doesn't make sense because the answer span is in them), but isn't this loss clearly in favor of DensePhrases? DPR estimates the similarity between the passage and the question, and the passage without the answer sentence is probably still relevant after removing the sentence with the answer. In contrast, phrase-based passage scores are based on the phrases in the passage, so removing the answer should help. A less biased alternative for hard negatives passages would be to use, for example, passages that get high scores by BM-25 but do not contain the answer.


### Questions

- Did you try using DensePhrases with $\mathcal{L}_{\text{doc}}$ for the QA tasks?

- Figure 3 - What are possible explanations for the higher topical loss of DensePhrases in comparison to DPR? Is it because the passage is ranked based on the scores of the phrases in it that were computed without context (the entire passage)?




### Suggestions

- Table 3 - a RAG baseline appears in the table but is not mentioned anywhere in the text, and the corresponding paper ("Retrieval-Augmented Generation for Knowledge-Intensive NLP Tasks", Lewis et al. NeurIPS, 2020) is not cited.

- "Passages and phrases are conceptually not so different" - this is arguable.

---

> ### Author Response · Authors · 2021-07-31
> **Thank you for your review!**
>
> Thanks for your comment. Following your suggestion, we updated the wording of the arguable sentence.
>
> Q1.  Isn't this loss ($\mathcal{L}_\text{hard}$) clearly in favor of DensePhrases? A less biased alternative for hard negative passages would be to use passages that get high scores by BM-25 but do not contain the answer.
> - It’s a valid point that current $\mathcal{L}_\text{hard}$ might favor DenePhrases. We also measured $\mathcal{L}_\text{hard}$ with BM25 negatives, but our phrase-based passage retrieval still had a better $\mathcal{L}_\text{hard}$ (0.27) compared to DPR (0.34) although DPR was explicitly trained on BM25 negative passages.
>
>
> Q2. Did you try using DensePhrases with $\mathcal{L}_\text{doc}$ for the QA tasks?
> - For open-domain QA, we tried using $\mathcal{L}_\text{doc}$ during query-side fine-tuning. We have found out that it can improve the passage retrieval accuracy, but it slightly sacrifices the end-to-end QA accuracy.
>
>
> Q3. What are possible explanations for the higher topical loss of DensePhrases in comparison to DPR?
> - Relatively higher $\mathcal{L}_\text{topic}$ of DensePhrases can be attributed to the following two facts: First, DensePhrases uses a smaller batch size (84) for the in-batch negatives compared to the batch size of DPR (256). This could be a disadvantage to DensePhrases when learning topical relevance. Second, since DensePhrases’s positive instances are phrases (not passages), it can be trained in a way that it only considers the local entailment between questions and phrases even in the presence of the in-batch negatives. Note that most of the topically different phrases can also be discriminated by their local cues since they will also have different local cues. In this situation, DensePhrases might lose a chance to encode topical information in their phrase representations.
>
> Thank you for your suggestions, which we’ve included in our revision.

---

### Official Review · Reviewer_eDvt · 2021-07-20
**Well written paper with clear empirical results**

**Rating:** 7
**Confidence:** 3

**Review:**

The paper shows that dense phrase Representations can be better representations for retrieval than the document-level representations. Specifically they show that DensePhrases representations can be used to identify relevant documents _without any fine-tuning_ on the target QA dataset compared to DPR. Additionally they show why phrase representations might outperform passage-level representations (much harder negatives). On non-QA datasets, fine-tuning the query representations can lead to competitive performance too. Finally they also provide a potential solution to reduce the index size of DensePhrases to match that of DPR while maintaining a higher performance.


# Strengths
---
 - A simple retrieval approach that outperforms current document-based representation model - DPR on QA and non-QA tasks
 - Allows for a trade-off between accuracy and index size easily
 - A new perspective on document retrieval that may lead to research in better retrieval approaches via phrase-based representation learning


# Weakness
---
- Some more analysis would be helpful


# Questions/Comments
---
- Phrase representations are designed to answer these questions directly so maybe it is not as surprising that the answer phrases (which are likely to be correct) are from the gold documents. How often does the DensePhrases model get the wrong answer but still retrieve a relevant document?

- One issue with such an approach would be the dependence on the definition of phrases. If the answer to a question is not extracted as a phrase, do you think it most likely would not be retrieved by such an approach?

- I see that using DensePhrases reduces the need for feeding long documents to FiD. However, the training resource comparison doesn't seem fair since you reduced the batch size. Couldn't the same hardware be used by Izacard & Grave with the reduced batch size (for the same k)?

- Table 3: Can you add a couple of words regarding RAG? Also it seems unclear why you mention that $L_doc$ performs comparably well with DPR-based models? There seems to be a huge gap between Multi-Task DPR and DensePhrases?

- A recent work that also converts documents into multi-vector representations and also similarly trades off accuracy with efficiency - "ReadOnce Transformers: Reusable Representations of Text for Transformers". I think their approach is sufficiently different and not comparable, but would be worth contrasting against.

- I like the analysis to show the impact on input document size and index size. What is the cost of using this representation on the inference time as compared to DPR? If each document has about k phrases, it would be k times slower than DPR (not accounting for the slowdown due to a larger index)? Also does the quantization to reduce the index size further slow down retrieval time?

---

> ### Author Response · Authors · 2021-07-31
> **Thank you for your review!**
>
> Thanks for your comment. We’ve revised the paper with a description of RAG.
>
> Q1. How often does the DensePhrases model get the wrong answer but still retrieve a relevant document?
> - If the definition of relevancy is based on whether the answer is included or not, it is straightforward to compute the proportion of questions having relevant passages with incorrect phrase-level answers at top-k: (top-k passage retrieval accuracy) - (top-k end-to-end QA accuracy). For the top-1 results of DensePhrases-multi, this corresponds to 51.1% - 41.3% = 9.8%.
>
> Q2. If the answer to a question is not extracted as a phrase, do you think it most likely would not be retrieved by such an approach?
> - We think that high-quality contextualized phrase representations will contain many salient features about the text: For instance, for a question “Why is the sky blue?,” our method retrieves a passage containing the sentence “The clear daytime sky and the deep sea appear blue because of an optical effect known as Rayleigh scattering” based on the phrase “Rayleigh scattering”. A generative reader model can then synthesize a longer answer, e.g.. “The sky is blue due to an optical effect known as Rayleigh scattering”. For non-factoid retrieval, we would point to our WoW dialogue-generation results.
>
> Q3. Couldn't the same hardware be used by Izacard & Grave with the reduced batch size
> - We used the same batch size (due to the resource constraint, we had to use gradient accumulation) so this is a fair comparison. We have revised  Table 2 to make it clearer.
>
> Q4.  What is the cost of using this representation on the inference time as compared to DPR?  Also does the quantization to reduce the index size further slow down retrieval time?
> - As both DPR and DensePhrases can be defined as a single MIPS search, a theoretical drawback of DensePhrases is the larger index size. However, we combat this with our quantization methods. In practice, retrieval speed can be very implementation-dependent and we find that DensePhrases runs comparably fast as DPR for passage retrieval. The only additional operation in phrase-based passage retrieval is to merge the top-k’ phrases into top-k passages, but this has a very small cost. Product quantization can actually further speed up the retrieval time, see Jégou et al., 2011.
>
> [1] Jégou et al., 2011. Product quantization for nearest neighbor search.

---

### Official Review · Reviewer_XsjV · 2021-07-21
**Interesting exploration of phrase retrieval**

**Rating:** 7
**Confidence:** 3

**Review:**

This paper explores how dense phrase retrieval (i.e., treating a phrase as the unit of retrieval) compares to dense passage retrieval in the context of QA and specialized document retrieval tasks. To improve phrase retrieval, the authors propose the use of in-passage negatives (i.e., negative phrases taken from a relevant passage) and discuss how this relates to strategies for choosing negative exmaples with passage retrieval. In the evaluation, the proposed approach is more effective than DPR (passage retrieval) for QA and competitive on the other tasks. Additionally, the authors demonstrate that quantization can be used to substantially decrease the phrase index size, which mitigates this potential downside of phrase retrieval.


# Strengths

This work demonstrates that a straightforward approach treating phrases as the unit of retrieval can be effective, which is an interesting perspective that can motivate future work. It additionally proposes a new strategy for identifying hard negatives in this setting and demonstrates techniques for mitigating the increase in index size that comes with this setting.

# Weaknesses

- The proposed approach does not always outperform RAG (which could be described in more detail) in the experiments.

- The reader is left to follow prior work for many basic details about the approach, such as exactly how a phrase is defined (how are they chosen? how long are they? can phrases overlap?) and how multiple phrases are used to score a document. (Eq 3 indicates that only the max score is used, but this is at odds with later statements about multiple representations.)


# Other comments/Questions

- How dependent is this approach on the definition of a phrase? Were other definitions considered?

- It's surprising that a phrase representation could include enough information to match it against a query with no lexical overlap. Do you see this approach as applicable to the general task of ad hoc document ranking? Could it be benefitting from the specifics of the datasets or tasks considered? Any insights or discussion along these lines would be very interesting.

---

> ### Author Response · Authors · 2021-07-31
> **Thank you for your review!**
>
> Thanks for your comment. We’ve revised the paper with a description of RAG in Section 5 - Results and more details on the definition phrase in Section 2.
>
> Q1. How dependent is this approach on the definition of a phrase? Were other definitions considered?
> - We think our definition of phrases is very general (Footnote 1), since we take a phrase to be any contiguous text segment up to L words. In future work, it would be interesting to constrain the definition to linguistic phrases.
>
>
> Q2. Do you see this approach as applicable to the general task of ad hoc document ranking?
> - Yes, we are currently planning to evaluate our model on ad-hoc document retrieval tasks like MS MARCO.

---

### Author Response · Authors · 2021-07-31
**General response**

We thank all the reviewers for their thorough comments. All three reviewers asked for a more detailed description of the baseline models in our KILT results, especially RAG (Lewis et al., 2020). We’ve added this in our revision. Although their method uses a generative reader model unlike other models in the table, we only compare the performance of the retrieval components.

---

### Decision · Program_Chairs · 2021-08-18

**Decision:**

Accept

**Comment:**

This paper presents a method for adapting phrase embedding and phrase retrieval techniques to be used for passage retrieval. Results show that DensePhrases can lead to better retrieval for open-domain QA than passage-level DPR, better open-domain QA performance, and generally better results on two other tasks from KILT. The reviewers overall found the paper very well-written and the ideas in this paper to be worthy: the results are convincing and future work can easily follow on this straightforward method. Concerns were chiefly about lack of description of the baselines, some other work the model could be situated with respect to, a perception that the analysis could be a bit deeper, and a lack of a concrete definition of phrase (although this is derived from prior work). However, these are minor issues and have been largely addressed in the responses.